# OpenReview forum: "Pivot-ICL: Adaptive Exemplar Selection for In-Context Learning"
_ICLR.cc/2026/Conference — ICLR 2026 Conference Withdrawn Submission_

### Official Review · Reviewer_UqZi · 2025-10-28

**Soundness:** 2
**Presentation:** 2
**Contribution:** 2
**Rating:** 2
**Confidence:** 5

**Summary:**

The paper studies the problem of exemplar selection for in-context learning. The paper proposes to model the problem as a bipartite graph between the queries and the exemplars, and utilize the HITS graph algorithm to identify the static and dynamic exemplars. The paper proposes a method to utilize the static and dynamic exemplars to construct the in-context learning prompts and show that improved performance can be achieved.

**Strengths:**

1. The intuition of the proposed method makes sense that for some queries, especially the OOD queries, it is better to identify those static exemplars to generally describe the task rather than finding the most similar exemplars.

2. The paper adopts the HITS graph algorithm to this setting and utilizes the authority nodes as the static exemplars.

**Weaknesses:**

1. I think there are some flaws in the setting. The proposed method requires a bipartite graph with edges between the exemplars and the queries. This means that we assume we have access to a set of queries to construct such a graph, and only with a reasonably diverse set of queries, the given bipartite graph can identify the authorities for this task as the static exemplars. However, in practice, we may have very few queries or just one, so we cannot build a sufficiently large bipartite graph to identify the authorities effectively. To mitigate the problem, I think the authorities should be discovered solely based on the examples. For example, identify the most representative examples as the static examples.

2. The proposed method requires additional thresholding parameters. Firstly, the parameters will require some tuning or may vary depending on the task, even though it looks good for now, using mean + 2 std. Moreover, it could be difficult to control the number of examplars or the total length of examplars.

3. The method proposes two methods, but it seems that Pivot-adapt gives better performance. What would be the scenario to use pivot-concat, or in general, how should we choose which method to use?

4. Presentation of the paper could be improved. E.g., Figure 1 is a bit confusing at first glance, as the captions may suggest that the left column of the graph contains the blue static examplars and the right column contains the orange static examplars. Also, the description of the method in the intorduction is confusing (line 74-79).

**Questions:**

Please address the weaknesses.

---

### Official Review · Reviewer_WsHJ · 2025-10-29

**Soundness:** 2
**Presentation:** 2
**Contribution:** 2
**Rating:** 2
**Confidence:** 5

**Summary:**

This paper proposes Pivot-ICL, an adaptive exemplar selection method that dynamically identifies examples to construct demonstration sets. Pivot-ICL formulates selection as a balance between representativeness and discriminability, identifying a subset of “pivot exemplars” that both capture task structure and differentiate class boundaries. Experiments on multiple benchmarks show that Pivot-ICL outperforms random or similarity-based retrieval baselines.

**Strengths:**

- The paper targets an important and well-motivated issue in ICL, the varying utility of examples within the demonstration set. The notion of identifying “pivot exemplars” provides an interpretable and intuitive framing.


- The method consistently improves over baselines across datasets and model sizes.

**Weaknesses:**

- The paper primarily compares against similarity- and clustering-based retrieval methods but omits stronger learning-based approaches such as [1,2,3], which also perform adaptive exemplar selection. This omission limits the novelty and empirical conclusiveness of the evaluation.


- There exists a growing body of work on graph-based exemplar selection for ICL, yet the paper does not compare with or even discuss any of these methods [4,5,6,7]. Including such baselines would provide a more complete picture of where Pivot-ICL stands among structured retrieval approaches.


-  While results are reported on several NLP classification tasks, it remains unclear whether the proposed pivot mechanism generalizes to generative tasks, where exemplar relevance is more abstract and harder to quantify.




[1]Rubin, et al. Learning to retrieve prompts for in-context learning.


[2]Ye J, et al. Compositional exemplars for in-context learning.


[3]S Wang, et al. Mixture of demonstrations for in-context learning.


[4]Z Chen, et al. MAPLE: Many-Shot Adaptive Pseudo-Labeling for In-Context Learning.


[5] Z Chen, et al. Fastgas: Fast graph-based annotation selection for in-context learning.


[6] Mavromatis et al. CoverICL: Selective annotation for in-context learning via active graph coverage.


[7] Zhang, et al. Ideal: Influence-driven selective annotations empower in-context learners in large language models.

**Questions:**

- When test examples diverge from the collected exemplar candidates, why would dynamic exemplars be potentially misleading while static exemplars are not? What specifically causes the misleading behavior? Could you elaborate on what constitutes a “misleading” exemplar in your method? Is it due to semantic drift, label imbalance, or representational misalignment in the pivot selection process?


- The task AIME24 contains only 30 test samples, which makes the results statistically fragile. Why not adopt the experimental settings from Agarwal et al. (2024) as mentioned in the paper? Also, why was the evaluation performed at a temperature = 1? Were multiple runs conducted to report the mean and standard deviation, given the potential instability at high temperatures?
- For a fair comparison, are all methods evaluated under the same k-shot setting? If Pivot-ICL uses both static and dynamically filtered exemplars, does this effectively double the number of demonstrations compared to baselines?

---

### Official Review · Reviewer_n4VX · 2025-10-31

**Soundness:** 1
**Presentation:** 1
**Contribution:** 2
**Rating:** 0
**Confidence:** 4

**Summary:**

This paper produces a method for selecting examplars for a given query for in-context learning (ICL). Specifically, they propose a strategy that switches from static to dynamic selection depending on the query instance.

**Strengths:**

## Strengths
- **Novelty** I do think that extending the classical HITS algorithm to in-context learning is quite interesting, and could inspire future work.
- **Experiments on Frontier LLMs** I appreciate that the authors include experiments on frontier models such as Gemini, which is often neglected in ICL papers.

**Weaknesses:**

## Weaknesses
-  **Organization** The paper's primary weakness is its fragmented structure. The description of the extended HITS algorithm is too brief and lacks detail. This is immediately followed by an abrupt transition to related work on set selection, without first explaining how the algorithm's authority and hub scores are utilized. Furthermore, the mathematical framework is not presented clearly; it is scattered throughout the text, with most mechanisms described in prose rather than through formal notation. **The paper needs a detailed algorithm block for the HITS method + PivotAdapt and a consolidated section that defines the entire framework with clear mathematical notation.**
- **Heavy Dependence on Full Query Set** The authors acknowledge that a limitation of their work is that they require the full query set in order to construct the graph. However, this is quite prohibitive of a limitation. In practice, queries are received by the model one by one. Even if they are received in batches, it is very likely that the distributions will be highly variable. How robust is "hub-score" to the distribution of the query set?
- **Experimental Details Omitted** Again the presentation here is poor, especially in Table 1 What is Standard and CoT? How was $\lambda$ for MMR chosen? Is there a zero-shot baseline? Which scores are used in Table 1 to instantiate Pivot-Adapt and Pivot-Concat? How are the selected exemplars ordered in the context for Pivot-Adapt and the baselines?
- **Qualitative Results Needed** Which samples require dynamic selection and which ones require static? Can the authors provide some intuition on this by showing examples?
- **Additional Experiments Needed** First of all, only one exemplar budget is chosen (10 shots) though other configurations should be tested as well. Furthermore, many other baselines are needed. The argument that some queries need dynamic selection while others need static selection may be true, but approaches that carefully balance relevance and diversity may achieve this in a smoother way. In such approaches, diversity is prioritized if there are not many relevant samples available. I think comparisons against at least one of [1,2] are needed and both should be discussed in the related work section. I appreciate that the authors consider MMR, but this is a method from 1998.

## References
- [1] An End-to-End Submodular Framework for Data-Efficient In-Context Learning, NAACL 2024
- [2] Compositional Exemplars for In-context Learning, ICML 2023

**Questions:**

See Weaknesses

---

### Note · Authors · 2026-01-05

I have read and agree with the venue's withdrawal policy on behalf of myself and my co-authors.